# Perspective Charts

Mia MacTavish*      Katayoon Etemad†      Faramarz Samavati‡      Wesley Willett§

University of Calgary

## ABSTRACT

We introduce three novel data visualizations, called *perspective charts*, based on the concept of size constancy in linear perspective projection. Bar charts are a popular and commonly used tool for the interpretation of datasets, however, representing datasets with multi-scale variation is challenging in a bar chart due to limitations in viewing space. Each of our designs focuses on the static representation of datasets with large ranges with respect to important variations in the data. Through a user study, we measure the effectiveness of our designs for representing these datasets in comparison to traditional methods, such as a standard bar chart or a broken-axis bar chart, and state-of-the-art methods, such as a scale-stack bar chart. The evaluation reveals that our designs allow pieces of data to be visually compared at a level of accuracy similar to traditional visualizations. Our designs demonstrate advantages when compared to state-of-the-art visualizations designed to represent datasets with large outliers.

**Index Terms:** Human-centered computing—Visualization—Visualization techniques—Information Visualization;

## 1 INTRODUCTION

Today we are faced with large amounts of data with varying complexity [25]. This makes the visualization of large datasets challenging, especially when viewing space is limited. Different tools and charts are suited to different types of data [16]. Bar charts are one of the most commonly used data visualizations as they are simple and easy to interpret. However, datasets with a large range, with important variation at multiple scales, present unique visualization challenges. Examples can commonly be found in population data, as illustrated in Figure 1, which shows population data for several Canadian cities. A vertical limitation of the viewing space may require that a large amount of compression be applied to the data, which makes differences between values less readable. For example, in Figure 1, the largest value is the population of Toronto; the scale of the chart needs to be set to accommodate such large values. Showing Toronto's population in the same chart as smaller cities such as Guelph and Kingston makes it difficult to measure the population of the smaller cities. When the scaling factor increases, or when data becomes more compressed, it becomes more difficult to make comparisons between pieces of data with close values.

The limitation that we focus on is the readability of charts with multi-scale variation in the dataset. A linear mapping between the range of the data and the height of the viewing space may result in undesirable compression of the charts. One potential solution is to use a non-linear mapping, such as a logarithmic function (see Figure 1, right). However, this type of mapping is difficult to read and understand in comparison to simple linear mappings [9].

---

*e-mail: mmactavi@ucalgary.ca

†e-mail: ketemad@ucalgary.ca

‡e-mail: samavati@ucalgary.ca

§e-mail: wesley.willett@ucalgary.ca

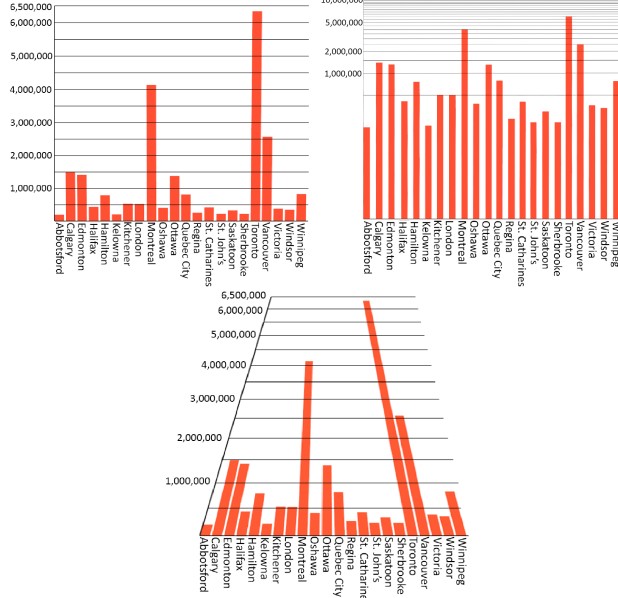

Figure 1: Canadian cities with a population of more than 150,000, in a *traditional bar chart* (left), a bar chart with a logarithmic scale (right), and a *slanted perspective chart* (bottom).

How do we find a more natural solution to mapping datasets with large outliers onto a small viewing space? We propose a new technique for visualizing data with important variation at multiple scales using perspective projection. Humans naturally perceive perspective, and are able to estimate the size of distant objects through a property known as size constancy [2]. Using simple linear perspective, geometric proportions can be used to measure the size and relative differences of objects [4].

Our first design, which we call the *slanted perspective chart*, shows a bar chart that is slanted backwards from the viewing plane, such that it is viewed in perspective (See Figure 1, bottom). As the lower part of the graph appears closer to the reader, small values in the dataset become larger in comparison to a *traditional bar chart*.

The main problem with the solution of slanting a *traditional bar chart* is that larger values in the dataset become compressed due to the perspective projection. This may make large values more difficult to read and compare.

Our next chart, the *stepped perspective chart*, is designed to address the issue of scaling large values in our *slanted perspective chart*, while also improving the readability of small values. In bar charts, space in some parts of the chart are often wasted due to large differences in values or outliers in the dataset. We can reduce the amount of wasted space by visualizing this area in an extreme slant. This puts only the less important range of the data at an extreme angle; each bar's value is still measurable in an area that is perpendicular to the view (see Figure 2, left).

This design is intended to resemble a staircase; we can insert multiple bends in the axis in a single chart to compress multiple

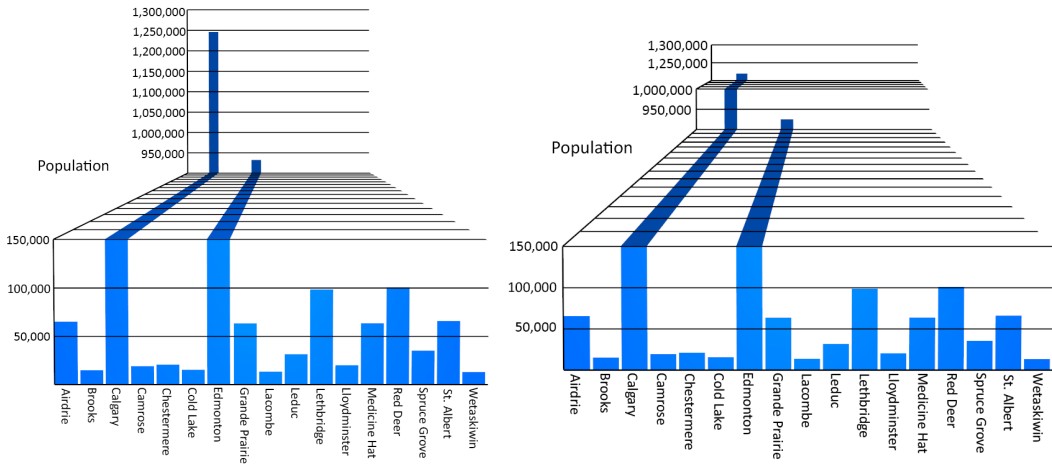

Figure 2: Left: A *stepped perspective chart*. Right: A *stepped perspective chart* with multiple bends in the axis.

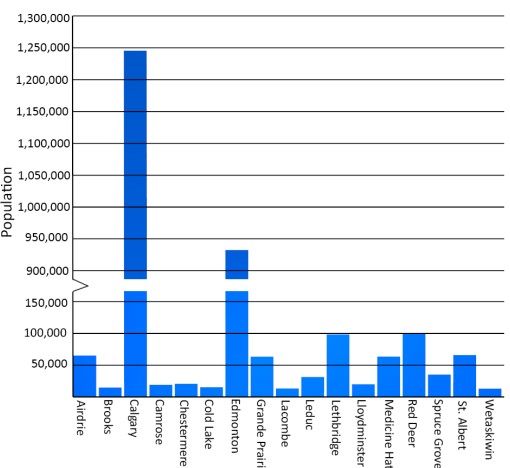

Figure 3: A broken-axis bar chart.

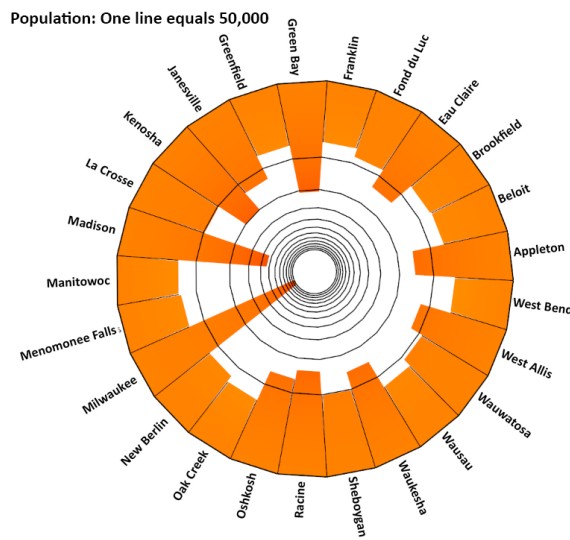

Figure 4: Population of cities in the state of Wisconsin in the United States, in a *circular perspective chart*. Each line marks an increment of 50,000.

areas of the chart and eliminate multiple areas of unused space (see Figure 2, right). Since the tops of the bars are not slanted or foreshortened in the *stepped perspective chart*, the values are emphasized more strongly than in the *slanted perspective chart*.

The *stepped perspective chart* is conceptually similar to a traditional *broken-axis bar chart* (see Figure 3), which also addresses the issue of wasted space in areas where there are large gaps in the data. However, since a *broken-axis bar chart* essentially cuts out a portion of the graph, the ability to visually estimate and compare data is lost, unlike in our *stepped perspective chart*.

Both our *slanted perspective chart* and *stepped perspective chart* contain some wasted space around the upper corners of the viewing space. To eliminate areas of unused space wherever possible, we introduce a third type of Perspective Chart, called the *circular perspective chart* (see Figure 4). Our design for this chart is inspired by the impression of looking up at tall buildings and skyscrapers from a low vantage point. The horizontal axis of the chart is mapped to a circle, with the vertical axis extending away from the reader's view. This chart occupies a consistent viewing space regardless of the scale of the data or the number of entries in the dataset.

The data visualization challenges that we discuss related to readability in datasets with multi-scale variation can be addressed using dynamic visualization methods, such as focus-plus-context; however,

we focus on a static method of addressing these issues. We introduce a new class of charts comparable to traditional static bar charts, and note that commonly used interactive techniques for bar charts can also be used with our *perspective charts*.

To evaluate our visualizations, we conducted a user study with twenty-four participants. The study quantitatively measured the speed and accuracy with which users could read data from our charts in comparison to traditional methods, such as a standard bar chart or a *broken-axis bar chart*, and state-of-the-art methods, such as a *scale-stack bar chart*. We also performed a qualitative evaluation of our three designs. Participants generally responded favorably to our visualizations, and were able to read data from them as accurately as with the traditional methods in fifteen out of seventeen task types, and performed more strongly than a recent method, *scale-stack bar charts* [9], in three out of four tasks.

## 2 BACKGROUND AND RELATED WORK

Given the increasing size and complexity of available datasets, finding clear and readable methods of visualization is becoming more

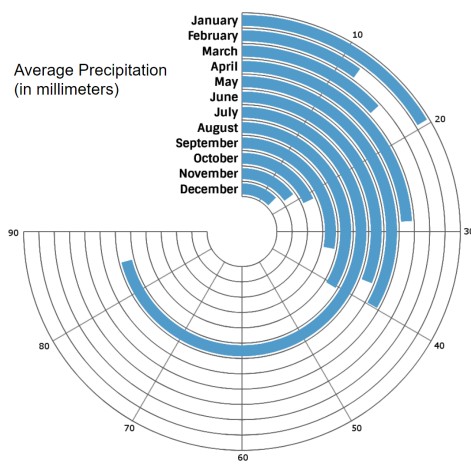

Figure 5: A *radial bar chart*.

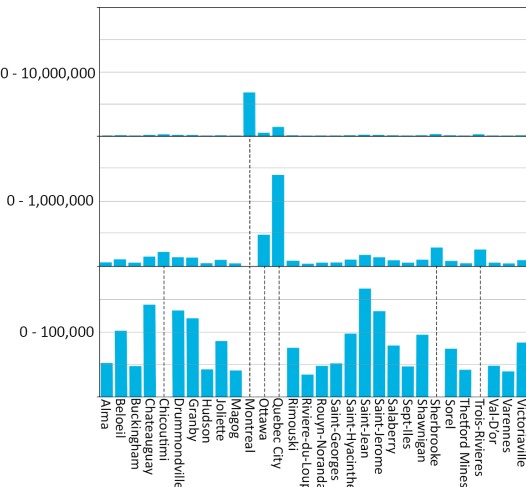

Figure 6: A *scale-stack bar chart*. The chart represents one dataset at three different scales stacked on top of one another.

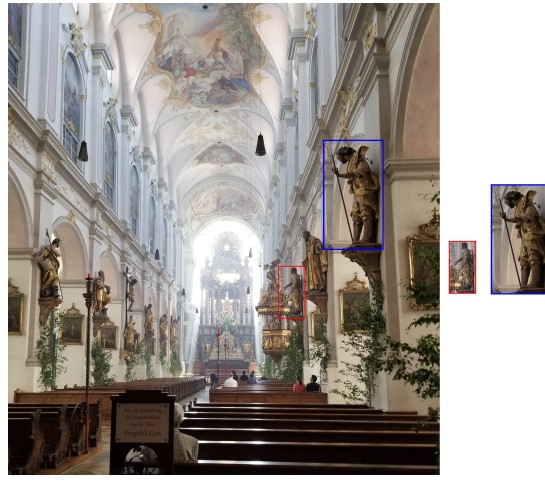

Figure 7: An example of single-point perspective. Due to size constancy, we perceive the height of the statues in the red and blue boxes to be equal. In image space, the statue in the red box is half the height of the statue in the blue box.

challenging [12]. Traditional methods such as bar charts are not always a practical choice when visualizing datasets with a large range with respect to important variations in the data [11]. In this section, we first provide a short review of research on visualizing complex datasets using variations of bar charts. Since we use perspective projection in our charts, we provide a short review of literature on the ways that perspective affects human perception.

## 2.1  Visualizing Data with Multi-Scale Variation

When a range of data is mapped to a bar chart, a scaling factor is applied such that all of the data can be represented in the viewing space. In datasets with large outliers, it may not be possible to fit the chart into a limited viewing space without applying scaling that decreases legibility. To address this problem, alternatives to bar charts are used in some applications. Karduni et al. wrap large bars over a certain threshold back over the y-axis in their Du Bois wrapped bar chart [10]; a similar technique is described by Reijner's horizon graphs [20] for time-series data, evaluated by Heer et al. [8] as an effective technique. Hlawatsch et al. compare their scale-stack bar charts with logarithmic and broken bar charts for the visualization of datasets with a large scale [9]. An example of a scale-stack bar chart is shown in Figure 6.

One traditional alternative to bar charts for this use case is the broken-axis bar chart. Broken-axis bar charts eliminate areas of unused space between values in a bar chart, visualized as a discrete jump in values in the y-axis. However, truncating the y-axis of a chart in this manner has been shown to negatively affect the perception of scale in datasets [3]. We compare broken-axis bar charts to our *stepped perspective chart* in our evaluation.

Charts scaled with a logarithmic function are also sometimes used to represent datasets with a large range. However, this type of scale is not typically used in bar charts, as it may be difficult to interpret given that it is non-linear [9].

## 2.2  Variations of Bar Charts

There exist several proposed solutions to common problems with bar charts. In cases where a guaranteed 1:1 aspect ratio may be desirable for a visualization, a circular chart such as a radial bar chart may be suitable (see Figure 5). A radial bar chart occupies a fixed viewing space regardless of the scale of its data. Luboschik's work on particle-based map label placement [14] highlights the use of circular charts in geospatial data visualization, where point-based icons are useful. Despite their popularity, circular chart types are generally discouraged by visualization experts, as they tend to be more difficult to read than a traditional bar chart [7].

Skau et al. evaluate the impact of visual embellishments in bar charts [22], taking into account human perception and aesthetic factors in their analysis. The results of their evaluation show that simple embellishments like rounded or triangular bars have strong effects on human perception, and in some tasks will negatively affect performance. Their evaluation found that humans rely on strong lines at the ends of the bars to accurately estimate values. In our *stepped perspective charts*, the tops of the bars remain visible and perpendicular to the view in each cluster of data.

## 2.3  Human Perception

Perspective, in combination with lighting, distance and angle, contribute to human perception of information [6]. The visual shape and size of objects changes as the object's distance and orientation changes relative to the viewer [21]. However, the concept of *size constancy* explains that the perception of an object's size does not change with the object's distance from the viewer [19]. This is true even for two-dimensional representations of three-dimensional scenes (see Figure 7). This is due to humans' natural ability to

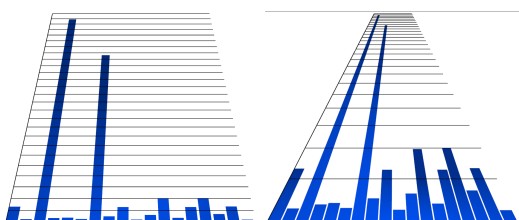

Figure 8: A comparison of methods for slanting a *traditional bar chart*. The left chart's vertical axis is slanted away from the viewer at an angle of $30°$, stretching the axis in the process. The chart on the right is slanted at an angle of $60°$.

account for perspective and the reduction of the projected size of an object when estimating its true size [24]. Size constancy is one of the types of natural constancy in human perception of distance and scale of objects. We use this feature of perception in the design of our *perspective charts*.

The use of 3D in data visualization, however, is controversial. Norman et al. found that visual perception of length was less accurate in 3D space than in a two-dimensional visualization [18]. Munzner et al. discourage the use of 3D in data visualizations, citing concerns with occlusion, distortion, and legibility of text [16]. Our *perspective charts* are carefully designed to avoid occlusion, and text is always displayed perpendicular to the viewer. 3D visualization techniques warrant further evaluation, with thoughtful design to address these common concerns.

Mackinlay et al. [15] use perspective projection in their technique called the Perspective Wall. This interactive technique addresses data with "wide aspect ratios" by placing the area of focus on a flat plane, with surrounding contextual data placed on planes slanted away from the viewer. Other aspects of human perception are used in the design of various hierarchical data visualizations, such as those described by Gestalt psychology principles [13] of closure [17] and continuity [5].

## 3 METHODOLOGY

We propose the use of perspective as a mapping of bar charts in three different designs: the *slanted*, *stepped*, and *circular perspective charts*. In this section, we present design rationale and methods for creating our three different *perspective charts*.

### 3.1 Slanted Perspective Charts

*Slanted perspective charts*, as shown in Figure 1, are similar to *traditional bar charts*, but have the vertical axis of the chart slanting away from the viewer. This design is inspired by drawings and images that portray one-point perspective, i.e. images with a single vanishing point, as shown in the photograph in Figure 7. We use a simple 3D environment and set a predefined camera setup to avoid user input for 3D interaction. Slanting the chart brings smaller values closer to the viewer, while moving larger values away.

The slant in the vertical axis of the chart can be achieved either by viewing the chart from a lower angle, or by maintaining the same viewpoint and instead slanting the chart plane backwards from the viewer in three-dimensional space. We choose the latter option in order to maintain a consistent viewing space. Slanting the chart moves large values in the chart away from the viewer, and decreases the space between scale lines.

We restrict the foreshortening ratio in order to limit the compression of large values as they move away from the viewer. The foreshortening ratio measures how objects viewed at an angle appear to be shorter than their true measurement. We slant the y-axis of the chart at a fixed angle $\theta$, which controls the foreshortening ratio $f_r$

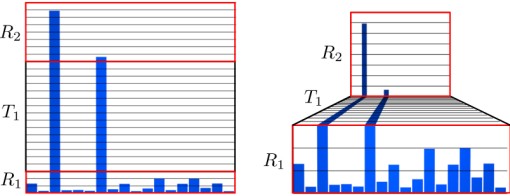

Figure 9: An example of a dataset with two clear clusters $R_1$ and $R_2$, and a transitive range $T_1$.

of the slanted line $L$ compared to the viewing plane $V$:

$$f_r = \frac{L}{V} = sec(\theta).$$

For example, when $\theta = 60°$, the foreshortening ratio $f_r$ is 2. In general, $1 \le f_r < \infty$ where $0 \le \theta < \frac{\pi}{2}$.

We avoid slanting the chart at extreme angles in order to control the foreshortening ratio $f_r$ and maintain readability of large values. Figure 8 demonstrates how a change in $\theta$ impacts foreshortening.

### 3.2 Stepped Perspective Charts

Our *stepped perspective chart*, as seen in Figure 2, resembles a staircase showing multiple "tiers" of data. According to these tiers, we divide the range of the data into subranges $R_1, T_1, R_2, T_2, ..., R_n$ (see Figure 9) where each $R_i$ is a cluster of the data and $T_i$ are transitions. Each of these subranges represents a rectangular region of the chart. To create the *stepped perspective chart* we use a vertical view plane with a view angle $\theta_v = 0°$ for the $R_i$, and an extreme slant ($\theta_v = 60°$ for Figure 9) for the transitive regions $T_i$.

The *stepped perspective chart* is comparable to traditional broken-axis bar charts, which are also intended to address issues associated with large gaps between values in a dataset. However, broken-axis bar charts have been shown to negatively affect the perception of scale in datasets [3]. In a traditional broken-axis bar chart, without the use of labels, it is impossible to visually compare values on opposing sides of the break in the axis. In the *stepped perspective chart*, the area within the gap is still visible, as it runs at a different angle rather than being cut out of the chart entirely (see Figure 9). This way, visual estimation is still possible, and the reader is able to perceive the approximate size of the gap. As in a *broken-axis bar chart*, values do not fall within the transitive region of the *stepped perspective chart*; this design is a modification of the common broken-axis technique for truncating a chart's y-axis.

The amount of the transitional region of the chart that is visible can be adjusted. Figure 10 shows varying heights from which the chart can be viewed. While the axes are always bent at an angle of $90°$, the height of the camera affects the view angle. A height that is too low results in a high view angle, with scale lines positioned so closely together that they are no longer readable, while a low view angle lessens the impact of the separate regions of the chart. We choose a height that is just high enough to allow the viewer to distinguish between scale lines. The exact appropriate height is dependent on the resolution and size at which the chart is viewed.

### 3.3 Circular Perspective Charts

As seen in Figure 4, our *circular perspective chart* is inspired by the perception of tall buildings as viewed from a low vantage point, converging on a singular vanishing point. In this chart, the bars are placed in a closed polygon and extend away from the viewer, converging at a vanishing point at the center of the polygon.

We create our *circular perspective chart* by bending the horizontal axis to remove areas of unused space and accommodate a larger number of values in a limited viewing space. We can imagine that

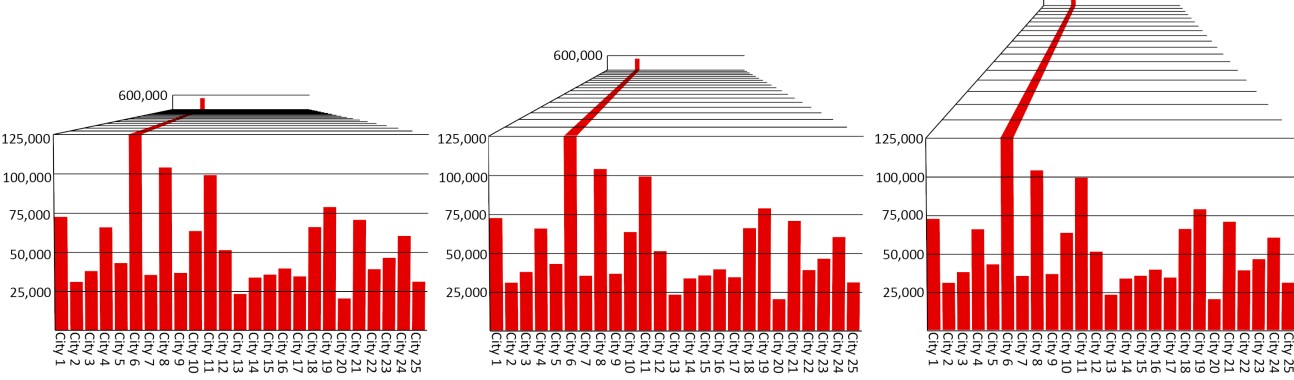

Figure 10: A *stepped perspective chart* viewed at three different heights, resulting in varying view angles and amounts of viewing space occupied by the transitional region of the chart. Left: $\theta_v = 80°$. Center: $\theta_v = 60°$. Right: $\theta_v = 40°$

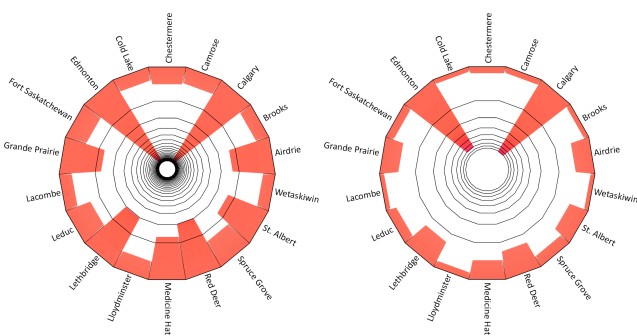

Figure 12: A *circular perspective chart* showing population of cities in Alberta, with two different scaling factors. In the top chart, each line marks an increment of 50,000. In the bottom chart, each line marks an increment of 150,000.

the entire bar chart is divided into multiple smaller sub-charts that are slanted individually (see Figure 11). The slanted sub-charts are then rotated to form a closed polygon. In the extreme case, each sub-chart is allocated to only one single value (see Figure 4). When there is no preferred clustering to create sub-charts, we use this extreme case as the main design of our *circular perspective chart*. By wrapping the horizontal axis of the chart to form a closed polygon, the chart is contained within a consistent view.

The vantage point of the viewer is a potential variable to use in interactive visualization using the *circular perspective chart*. Figure 12 shows an example of a low and a high viewing height for the chart shown in Figure 4. In the left chart, each scale line represents an increment of 50,000 for a total of twenty-four scale lines. The right chart's scale lines represent an increment of 150,000 for a total of eight scale lines. To make efficient use of space in the *circular perspective chart*, the scale should be chosen such that bars are compressed as little as possible while avoiding the issue of closely converging scale lines. The occurrence of this issue is dependent on the size and resolution at which the chart is displayed.

The use of circular chart types is generally discouraged by visualization professionals [7]. However, they are frequently used in practice due to their known 1:1 aspect ratio, independent of the number of bars represented. This property is also present in our *circular perspective chart*. Luboschik demonstrates that circular chart types are useful in geospatial data visualization [14]. The use of bar charts in spatial data visualization may present limitations in the available viewing space, hence it is worth exploring a circular chart type that has a consistent aspect ratio.

## 4 EVALUATION

We conducted a within-subjects user study to evaluate the readability and visual appeal of *perspective charts*. The study quantitatively measured the accuracy and speed with which users answered a series of questions based on data shown in various charts, and collected participants' opinions of the three types of *perspective charts* in a qualitative study.

In our user study, we evaluate the following hypotheses:

H1 For data with important variation at multiple scales, small values are more easily readable in a *slanted perspective chart* than in a *traditional bar chart*.

H2 The *stepped perspective chart* allows for faster and easier estimation and comparison of values than broken-axis bar charts and *scale-stack bar charts*, other axis-breaking methods for visualizing datasets with large outliers.

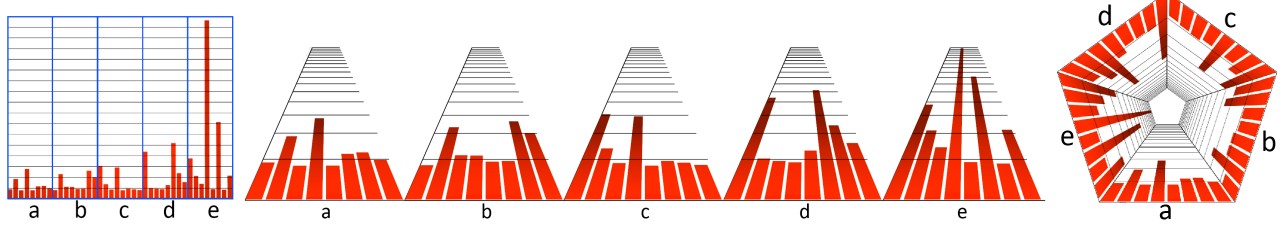

Figure 11: The *circular perspective chart* is created by grouping sections of a large chart into several smaller sub-charts. In this example we have five sub-charts. Each of the sub-charts is individually slanted, then rotated to form a closed polygon.

H3 In a dataset with important variation at multiple scales, small values are more easily readable in a *circular perspective chart* than in a *traditional bar chart*, while occupying a consistent viewing space. The ease-of-use of the chart should be comparable to existing fixed-viewing-space visualizations like the *radial bar chart*.

We describe one hypothesis per *perspective chart* design. The hypotheses are designed based on the concept of size constancy. In each block of tasks, smaller values, or values that appear "closer" to the user should be more easily readable due to the increased scale, without sacrificing readability of larger values, which appear farther away from the user and therefore visually smaller.

H1 compares a *traditional bar chart* to the *slanted perspective chart*, a simple modification of a traditional visualization that introduces three-dimensional perspective. H2 compares the *stepped perspective chart* to other chart types that use axis-breaking methods for showing important variation at multiple scales. Since the *stepped perspective chart* uses a bend in the axis to show a large difference in scale between values, we choose to compare this design to existing chart designs that feature breaks in the y-axis. We evaluate whether the ability to visualize the area within the axis break allows values on either side to be more easily compared. H3 compares the *circular perspective chart* to a *traditional bar chart* and a *radial bar chart*. This is to evaluate the *circular perspective chart*'s performance compared to an existing circular chart type as well as a traditional chart type.

## 4.1 Study Design

We performed studies on an individual basis over the course of approximately 60 minutes per participant. Each participant was shown a series of various types of charts and answered a list of questions based on the data in these charts. Each participant performed tasks based on common visualization task taxonomies [1, 23]. Participants answered questions based on a *traditional bar chart*, a *radial bar chart*, a *broken-axis bar chart*, a *scale-stack bar chart* [9], as shown in Figure 6, and our *slanted*, *stepped*, and *circular perspective charts*.

### 4.1.1 Tasks

We designed six types of tasks based on visualization taxonomies [1, 23]. The task types used are:

- Retrieve Value – *"What is the population of Franklin?"*

- Determine Magnitude Difference – *"How much larger is Milwaukee than Oak Creek? (For example, 2x larger, 3.5x larger)"*

- Determine Range – *"What is the range of the data? (Smallest population to largest population)"*

- Find Extremum – *"Which city has the smallest population?"*

- Filter – *"List all cities with a population of less than 100,000."*

- Sort – *"Sort the cities by population from smallest to largest."*

In our evaluation we describe two different types of tasks for determining the magnitude difference between values: between two small values, and between a small and a large value. For each dataset, we can identify its largest transitive region, as in Figure 9. For this task type, a value is considered "small" or "large" depending on which side of this region it falls.

### 4.1.2 Methodology

Each participant completed five task blocks (B1 - B5), during which they completed a set of tasks using one chart type followed by a matching set of tasks using a second chart type:

B1: *traditional bar chart / slanted perspective chart*

B2: *scale-stack bar chart / stepped perspective chart*

B3: *broken-axis bar chart / stepped perspective chart*

B4: *radial bar chart / circular perspective chart*

B5: *traditional bar chart / circular perspective chart*

Within each block, each participant first performed a series of either four or five different tasks (see Section 4.1.1) using a chart of the first type, then completed a matching set of tasks using a chart of the second type that visualized a different data set. We maintained the same block, task, and dataset order for all participants, but varied the order of the chart types within each block. For example, in B1 half of the participants completed their first five tasks using a *traditional bar chart*. The other half started by completing the same set of tasks using a *slanted perspective chart* that visualized the exact same data. By placing B2 before B3 in our groupings, we ensure that both the *scale-stack bar chart* and the *stepped perspective chart* are unfamiliar to users when they are compared against each other in B2.

We chose this blocking scheme for the evaluation in order to maintain a pairing between our designs and existing chart types, and tailor task types within the blocks based on the charts used.

Each participant completed a total of forty-two tasks – five tasks (determine magnitude difference between a small and a large value, determine magnitude difference between two small values, filter, find extremum, and retrieve small value) for each chart type in B1, and four tasks for each chart type in B2-5. B2-5 had three tasks of the same type (determine magnitude difference between a small and a large value, determine magnitude difference between two small values, and retrieve small value), and one additional task of a varying type. Participants completed a find extremum task for B2, a retrieving large value task for B3, a sorting task for B4 and a filtering task for B5.

After completing the main set of tasks, participants answered a short post-study questionnaire evaluating each type of chart used in the study. This was followed by a short verbal interview where we further gathered their opinions on the charts used in the study.

### 4.1.3 Participants

We recruited 24 participants (12 female) using posters distributed across our local campus as well as via word of mouth. We based this cohort size on those used in similar evaluations [9] [8]. Twenty-two participants were students at the time of the study; 16 participants studied in STEM fields, while the remaining participants worked or studied in arts, business or social sciences.

A post-hoc power analysis was performed on the results of our evaluation for each task type in each block with our sample size of 24 participants, using GPower 3.1. Among these analyses, the lowest reported power was 0.74. Thus for each task type in our evaluation we have at least a 74% probability of finding true significance, given our sample size of 24.

### 4.1.4 Datasets

Since tasks were divided into five different blocks for each participant, we used two unique datasets in each block for a total of ten unique datasets. We chose real datasets that satisfied our use case of clusters of data across a large range. Six datasets showed population data across various regions, two represented pollution data, and

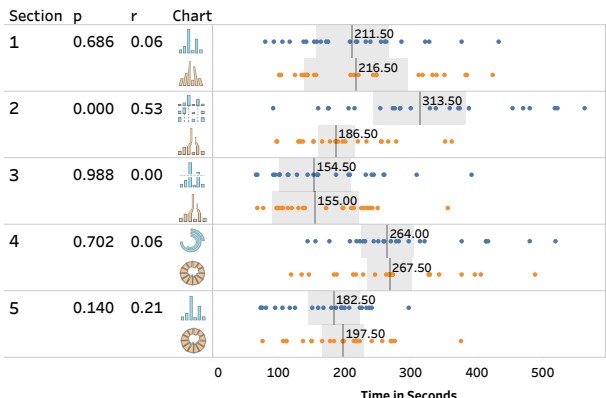

Figure 13: Results showing the total time for participants to complete all tasks in a block of the evaluation. For this and subsequent charts, the bar in the grey box represents the median error rate. The p-value and effect size (r) of each task type is shown in the table. Each box denotes the 95% confidence interval of the median. Each dot represents the task completion time of an individual participant.

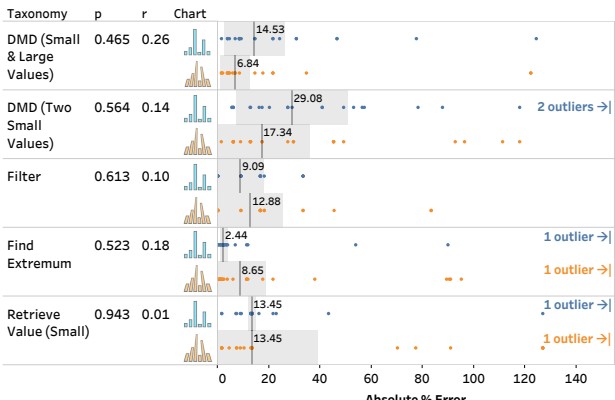

Figure 14: Accuracy results for block one of our evaluation, comparing the *slanted perspective chart* (orange) to a *traditional bar chart* (blue). Results are grouped by task type. Each dot represents the percentage of error of an individual response. For this and subsequent charts, each box denotes the 95% confidence interval of the median.

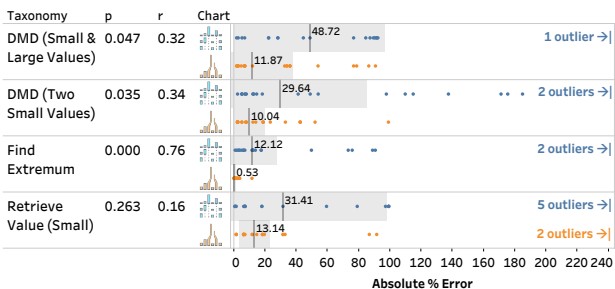

Figure 15: Accuracy results for block two, evaluating the *stepped perspective chart* compared to the *scale-stack bar chart*.

two represented precipitation data. We used data that was likely to be unfamiliar to the participants to reduce potential bias resulting from preexisting knowledge about the data. We did this by using data from regions that were not geographically close to the location where the study was performed, or by obscuring the names of the locations in the datasets ("City 1" instead of "Vancouver", etc.).

### 4.1.5 Test Environment

Before beginning the study, participants were given an explanation of the consent process, monetary compensation and risks associated with the study, as approved by the Conjoint Faculties Research Ethics Board of the University of Calgary. After indicating their consent, participants completed a short pre-questionnaire, then proceeded to the main portion of the study. Tasks were completed on paper in a prepared booklet provided to participants.

Each chart presented to participants included scale and axis labels in a consistent font style and size across charts. For the *circular perspective chart*, the chart's scale was labeled in the top-left corner of the visualization, as in Figure 4. Bars were unshaded to maintain simplicity in the chart designs. The appearance of charts used in the evaluation is comparable to the charts in Figures 1, 5 and 6.

### 4.2 Results

We compare task completion time and percentage of error between sets of tasks. The Shapiro-Wilk test indicates that our data does not follow a normal distribution, so we compare methods using a Mann-Whitney U test. As a result, we examine the median error rate for each task. For each test we report effect sizes (r) and p-values. All data is shared on the Open Science Framework (https://osf.io/w3fce/?view_only=23ff1dded0b74363a68ec86419c9c373).

### 4.2.1 Time

Timing data was measured as the sum of the total amount time each participant took to complete all the tasks for one block. Participants had the same number and type of tasks for each chart in a pairwise comparison. This data is shown in Figure 13.
***Slanted perspective chart:*** We did not observe a significant difference in task completion time ($p = 0.687$, $r = 0.06$) between the *slanted perspective chart* and a *traditional bar chart*.
***Stepped perspective chart:*** We observed that participants' task completion time was significantly faster ($p < 0.001$, $r = 0.53$) using our *stepped perspective chart* ($med = 186.5$ s) than a *scale-stack bar*

chart ($med = 313.5$ s). We did not observe a difference ($p = 0.988$, $r = 0.00$) between our *stepped perspective chart* and a *broken-axis bar chart*.
***Circular perspective chart:*** We did not observe a significant difference in task completion time ($p = 0.140$, $r = 0.21$) between our *circular perspective chart* and a *traditional bar chart*. There was also no significant difference ($p = 0.702$, $r = 0.06$) between our *circular perspective chart* and a *radial bar chart*.

### 4.2.2 Accuracy

We examine the absolute percentage of error when determining statistical significance of accuracy results; we compute this for each task by comparing each participant's numerical response with the true correct value.

For sorting tasks and filtering tasks, which had non-numerical responses, error rate was determined by counting the number of mistakes made compared to the true answer, and deducting "points" for each incorrect response. For example, in a filtering task to identify the number of cities with population below 100,000, if ten cities fulfill this criteria, a response that gives only nine correct cities would result in an error rate of $1/10 = 10\%$.

We did not observe a significant trend in the the directionality of error for any task block. For each block of tasks, results were corrected for multiple comparisons to control the false discovery rate. Median results for each block of the evaluation are shown in Figs. 14 to 18.
***Slanted perspective chart:*** Between this and a *traditional bar chart*, no significant difference was shown in the median error rate for any

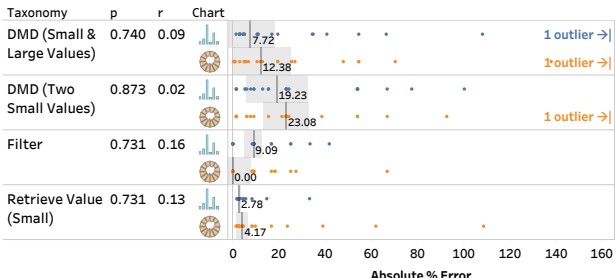

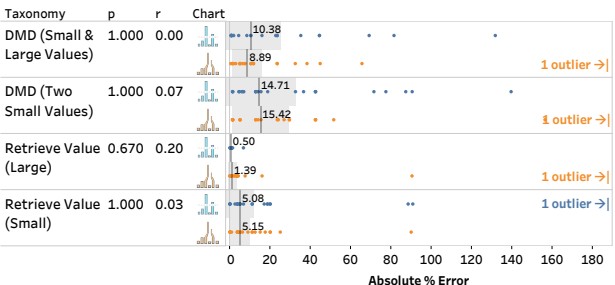

Figure 16: Accuracy results for block three, evaluating the *stepped perspective chart* compared to a broken-axis bar chart.

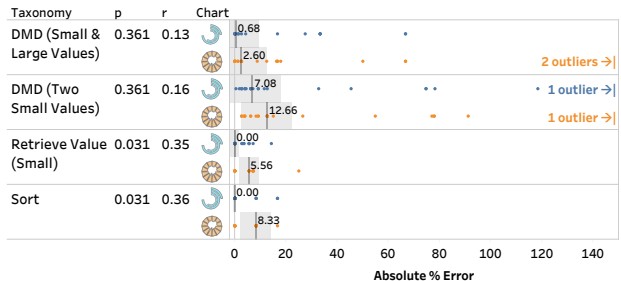

Figure 17: Accuracy results for block four, evaluating the *circular perspective chart* compared to a *radial bar chart*.

task type ($p > 0.4$, $r < 0.3$ for all tasks, see Figure 14).

***Stepped perspective chart:*** Between this and a broken-axis bar chart, there was no significant difference in accuracy demonstrated for any task type ($p > 0.6$, $r < 0.3$ for all tasks, see Figure 16).

The *stepped perspective chart* significantly outperformed the *scale-stack bar chart* for tasks related to determining magnitude difference between large and small values (*stepped med = 11.87*, *scale-stack med = 48.72*, $p = 0.047$, $r = 0.32$), determining magnitude difference between two small values (*stepped med = 10.04*, *scale-stack med = 29.64*, $p = 0.035$, $r = 0.34$), and finding extremum (*stepped med = 0.53*, *scale-stack med = 12.12*, $p < 0.001$, $r = 0.76$). We did not observe a significant difference in error rate for tasks related to retrieving small values from the chart ($p = 0.263$, $r = 0.17$). Results are shown in Figure 15.

***Circular perspective chart:*** No significant difference was shown in the median error rate for tasks performed with a *traditional bar chart* and our *circular perspective chart* ($p > 0.7$, $r < 0.2$ for all task types) (see Figure 18).

Compared to the *circular perspective chart*, participants were significantly more accurate using a *radial bar chart* for tasks related to retrieving values (*circular med = 5.56*, *radial med = 0.00*, $p = 0.031$, $r = 0.35$) and sorting (*circular med = 8.33*, *radial med = 0.00*, $p = 0.031$, $r = 0.36$). No significant difference was demonstrated for tasks determining magnitude difference, either between small and large values ($p = 0.361$, $r = 0.13$), or two small values ($p = 0.361$, $r = 0.16$). Results are shown in Figure 17.

### 4.2.3 Readability and Appeal

We gathered participants' opinions on the different types of visualizations in a post-study questionnaire. Responses are visible in Figure 19. The two most well-known types of visualizations, the *traditional bar chart* and the *broken-axis bar chart*, were the most well-received. The other types of visualizations were less familiar to participants and received lower scores, with the *circular perspective chart* receiving slightly less favourable scores.

Figure 18: Accuracy results for block five, evaluating the *circular perspective chart* compared to a *traditional bar chart*.

***Slanted perspective chart:*** According to the post-study questionnaire, the *slanted perspective chart* was the most well-received of our three designs. Two participants described the chart as "straightforward" (P1, P8). P12 stated that they preferred the *slanted perspective chart* because it allows the user to "see the whole range of data, unchanged, and still read the smaller values."

***Stepped perspective chart:*** Several participants indicated that they felt the design of the *scale-stack bar chart* was complicated, which made it more difficult to use. P10 felt it was "hard to go back and forth" between scales when using this chart.

***Circular perspective chart:*** Two participants felt that the *circular perspective chart* was more suitable as an artistic representation of data. P6 felt it was an effective chart type for "making an impact."

The *circular perspective chart* received the most "very hard to use" scores out of all the types of visualizations; nine out of twenty-four participants found it difficult to retrieve large values from the *circular perspective chart*. P10 noted that they would often "lose count when the perspective got smaller for the higher numbers." Nine out of twenty-four participants indicated that the effect of the perspective was too extreme in the *circular perspective chart*.

## 5 DISCUSSION

When compared to traditional chart types, we saw similar results for our three different designs; none of these comparisons showed a significant difference in accuracy rate for any of the evaluated task types. This suggests that due to size constancy, the use of perspective did not impact participants' ability to visually interpret values.

***Slanted perspective chart:*** While we initially hypothesized that reading small values would be easier for participants using a *slanted perspective chart* than in a *traditional bar chart*, the results did not demonstrate a significant difference in the accuracy or completion time of the two charts. Since our introduced charts are unfamiliar visualizations for the general public, it is unsurprising that participants were more comfortable with traditional methods, as indicated by the post-study evaluation and interview.

Participants performed similarly with both types of charts. This demonstrates that the use of perspective did not hinder their ability to perform tasks, due to size constancy. However, there was no evidence that participants retrieved small values more accurately with our designs as we hypothesized they would. In fact, median error rate was exactly the same for tasks of this type with both the *slanted perspective chart* and a *traditional bar chart*.

***Stepped perspective chart:*** As with the *slanted perspective chart*, the *stepped perspective chart* showed no significant difference in accuracy or timing compared to the traditional comparison method, in this case the *broken-axis bar chart*.

Participants were able to complete tasks significantly more quickly and accurately with the *stepped perspective chart* than the *scale-stack bar chart*, another unfamiliar visualization. This reinforces that axis-breaking methods have a negative affect on visual

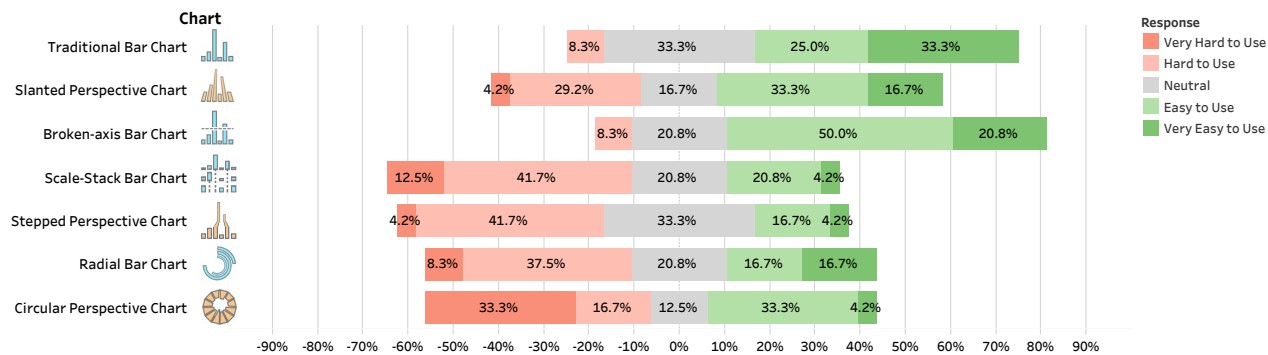

Figure 19: Results of our evaluation's post-study questionnaire.

perception, as suggested by Correll et al. [3]. These results also suggest that the use of perspective allowed for a more intuitive understanding of the visualization than the *scale-stack bar chart* for representing datasets with important variation at different scales. Charts that utilize perspective and size constancy may be a viable alternative to axis-breaking techniques for data visualization.

***Circular perspective chart:*** The *circular perspective chart* did not show a difference in accuracy compared to a *traditional bar chart*. However, the comparison fixed-viewing-space visualization, the *radial bar chart*, showed significantly higher accuracy than the *circular perspective chart* for sorting and value-retrieving tasks.

A previous evaluation performed by Goldberg and Helfman suggested that task completion speed was significantly lower in circular chart types than in traditional bar charts, due in part to the placement of labels relative to the chart's data [7]. One of the evaluated charts was a radial area graph, which has similar label placement features to the *circular perspective chart*. However, we have not observed a significant difference in task completion time between the *circular perspective chart* and either the *radial bar chart* or a traditional bar chart.

Participant feedback about the *circular perspective chart* was mixed but generally more negative than the other charts shown in the study; some participants felt that the chart was visually interesting but perhaps not suitable for retrieving data in the same way as the other evaluated charts. In the interview portion of the evaluation, several participants indicated that for larger values, the bars became too severely compressed in the *circular perspective chart*.

Based on error rates and participant feedback, it seems that the *circular perspective chart* design is not often suitable for accurate reading of values, as some participants stated the design was disorienting or confusing. However, participants also felt that the design was impactful, and may be appropriate for artistic visualizations.

While it is promising that participants had similar accuracy between our designs and traditional chart types, there were limitations to our evaluation. Further studies could evaluate the potential of size constancy applied to data visualization. Our evaluation has demonstrated that it is a viable solution to certain types of visualization challenges.

## 6 CONCLUSION

We have introduced three novel chart designs, called *perspective charts*, to address limitations of traditional bar charts caused by undesirable scaling factors in a fixed viewing space. Our designs can open up new possibilities for visualizing datasets with multiscale variation using the natural perception of size constancy. We provide design rationale for our three chart designs and evaluate their usability in a user study.

Evaluation showed no significant difference in performance between traditional visualizations, a *traditional bar chart* and a *broken-axis bar chart*, and our *slanted* and *stepped perspective charts*. This suggests that the use of perspective did not affect participants' ability to perform tasks in these types of charts.

Participants performed tasks significantly more quickly and accurately with the *stepped perspective chart* than with the *scale-stack bar chart*, another recent visualization design intended to represent datasets with important data at multiple scales, in three out of four task types. The *circular perspective chart* showed less accurate results than a *radial bar chart* for some tasks.

Our *circular perspective chart* was generally not well-received by participants. Some participants raised concerns about the compression applied to large values in the chart, and others expressed that it may be more suitable as an artistic data visualization.

Results for H1 and H2 indicate that the use of perspective projection in data visualization is worth further examination. The designs of the *slanted perspective chart* and *stepped perspective chart* were positively received by participants and performed comparatively to traditional methods in our evaluation. The *stepped perspective chart* in particular performed well compared to an existing visualization design, the *scale-stack bar chart*, for our use case of visualizing data with important variation at multiple scales.

Further evaluation could reinforce the results observed here. Based on our evaluation results, the slanted and stepped perspective charts particularly merit more in-depth evaluation as alternatives to existing chart types.

## 7 ACKNOWLEDGMENTS

The authors would like to thank Lora Oehlberg for her guidance with the analysis of our evaluation results.

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
