# OpenReview forum: "Perspective Charts"
_graphicsinterface.org/Graphics_Interface/2021/Conference/Second_Cycle — GI 2021_

### Official Review · Reviewer_8iVU · 2021-05-02
**Good study and analysis, but prospective bar charts have known perceptual problems**

**Rating:** 6
**Confidence:** 4

**Review:**

This paper presents a study comparing between traditional bar charts with variations of prospective chart in order to deal with datasets with large outliers. The study shows that the slanted and prospective charts does not necessarily improve the performance of users. Qualitative results suggest that prospective charts are harder to use than traditional charts.

Overall,  the study design is generally sound and the analysis is done well to draw careful conclusions. However, the main issue is that the proposed solutions of prospective charts are known to have problems from perceptual perspective. Therefore, it is not surprising that the performance did not improve over traditional charts and users find it more difficult to read these perspective charts. The problem of perception with 3D bar charts are well known (see 'visualization design and analysis' by Munzner, Ch 6). Unfortunately, the paper does not focus much on such issues in the related work section or elsewhere. There are some variations of traditional bar charts that may offer better solution than prospective bar charts in addressing large outliers, e.g. using a scale break as follows:

https://tomhopper.files.wordpress.com/2010/08/bar-chart-natural-axis-split1.png

This work would certainly strengthen from comparing with such simple variants of 2D bar charts that avoids the perceptual issues with 3D barcharts.

---

### Official Review · Reviewer_bNtb · 2021-05-03
**A decent study of some original visualization designs**

**Rating:** 6
**Confidence:** 5

**Review:**

The idea of perspective charts is interesting and original. The motivation makes sense (i.e. they seem well suited to dealing with datasets with large outliers, or with both small and large variations).

In the discussion of multi-scale variation techniques, I would expect a discussion of [a] given that, although this is for time series visualization rather than bar charts, the approach focuses on the same problem: that of having multi-scale variations. The technique seems particularly relevant to compare to the stepped perspective chart shown in Figure 2.
This brings me to the next question: why not comparing the proposed techniques to a "horizon bar chart", that would likely be a more fair comparison than say, the broken-axis bar chart or the scale-stack bar chart, which is clearly (and with good reasons) seldom used?

I am a bit confused by the statement "to evaluate the readability and novelty of perspective charts." -> how does one evaluate novelty?

There are some very good aspects to the evaluation section, like its overall good design, using meaningful, varied tasks, and the grouping of paired charts. I thank the authors for sharing their data on OSF and for providing carefully-crafted, informative figures.

On the other hand, the study appears to be testing way too many things. There are 3 hypotheses, 7 different types of charts, and 6 different tasks. This means that to keep the study under a manageable completion time, participants also performed each task only once for each chart, increasing the likelihood of getting noisy data than if using repetitions. Looking at the figures in the results section, the data points are indeed quite dispersed and the confidence intervals sometimes very large (e.g., an absolute error with a CI of [0,100%] is quite large). In addition, the complexity of the experiment with so many conditions and comparisons made probably justifies the use of an adjustment for multiple comparisons - which would enlarge even more the confidence intervals.

I do not see how novelty is measured in 4.2.3. Also, these are not qualitative results, but quantitative results with answers given on a scale.

Overall, this is one of these papers that introduces several design ideas that do not seem worth investigating at a first glance or at least with great skepticism, however, there is value in asking these kinds of questions. The results are not surprising: these designs that are difficult to read are indeed difficult to read. I am on the fence regarding the usefulness of such findings for the visualization community. Especially when the experiment designed to evaluate these designs seems too complex, leading to low experimental power and noisy data.

[a] Interactive horizon graphs: improving the compact visualization of multiple time series https://dl.acm.org/doi/10.1145/2470654.2466441

---

### Official Review · Reviewer_p9eP · 2021-05-04
**INteresting study on perspective visualizations**

**Rating:** 7
**Confidence:** 5

**Review:**

The authors report on a careful study of perceptive-projected visualizations, of which 2 use the bar chart form and 1 uses the radial form.  They hypothesized that the perceptual capability of size constancy would enable people to effectively decode these charts, particularly for small values. This is an interesting question, as visualization researchers for years have generally eschewed or actively denied the potential of perspective charts, given substantial research that suggests judgments based on perspective space are flawed. (The general argument is that size constancy holds for "real-world" objects but not necessarily for abstractions because we have highly contextual canonical shape knowledge of how objects behave in the real world.). In fact, the design principle not to use perspective charts is best articualted by Edward Tufte, who refers to the lie factor in perspective charts with respect to the data-ink ratio.   So any examination of whether - and how- this holds true is worth doing.   The authors found essentially no advantage for the perspective charts,  but claim few disadvantages, with the exception of the circular perspective chart.  They did however highlight distinct disadvantages of other approaches, notably the split or stacked chart.

The paper is overall interesting and the study useful, but there are some significant shortcomings that should be addressed.
Related Work:
- the lack of any reference to the famous Tufte design principle is a major flaw in their review of related work and should be addressed.
they need more background in the visual perception field to understand some of the artifacts that might occur. Norman, J. F., Todd, J. T., Perotti, V. J., & Tittle, J. S. (1996). The visual perception of three-dimensional length. Journal of Experimental Psychology: Human Perception and Performance, 22(1), 173–186. https://doi-org.proxy.lib.sfu.ca/10.1037/0096-1523.22.1.173 is a good reference.

Study/Stimulus Design
- Of some concern is the visualization design of the stepped bar chart, in which all the examples shown in the paper do not have any comparable values on the perspective (step) part of the chart.   If that is truly the case, then the authors are in sense cooking the books because comparing values on and off the stop is likely significantly harder and more error prone than values in the same place.
- why did they not counterbalance or at least randomize the order of the blocks? This can lead to first and second order learning effects.
- Their tasks were reasonable, but i wonder if they varied the proximity pf the two values they compared. This is a huge issue perceptually. If not, or if they did not control for it, they need to explain why. This is a large potential confound.

The paper is very well written, and I particualrly applaud the authors for the careful results figures, as they were nicely done and very readable.

---

### Meta-Review · Area_Chair_ietk · 2021-05-05

**Recommendation:** Accept
**Confidence:** 4

**Metareview:**

The three reviewers of this paper, all with high to very high expertise on the topic, agree that the paper should be accepted at GI.

In summary, the reviewers highlight the following strengths of the paper:
- addresses an interesting research question,
- proposes to provide empirical evidence to fuel the discussion around perspective charts,
- presents a well-designed study,
- is well written and with useful, well-crafted illustrations.

The reviewers make several good suggestions for small, but important improvements:
1. discuss Tufte's work and provide some more background in visual perception,
2. clarify the unclear points raised in individual reviews, including:
- better explain the choice of charts to compare to or not, including some other 2D chart techniques without perspective distortion also designed to deal with large outliers,
- explain the choice (and bias) to not have values on the perspective part of the step bar chart,
- explain the reasons for not counterbalancing/randomizing the order of blocks,
- describe and explain the potential confound of not varying the proximity of the two values being compared,
- discuss the (potential) problem of multiple comparisons in the analysis of the results,
- clarify whether participant data was aggregated per participant or not.

---

### Decision · Program_Chairs · 2021-05-08

Accept